

# Variation along liquid isomorphs of the driving force for crystallization

**Ulf R. Pedersen[1], Karolina Adrjanowicz[2,3], Kristine Niss[1] and Nicholas P. Bailey[1]**[*]

**1** Dept. of Science and Environment, Roskilde University, Roskilde, Denmark
**2** Institute of Physics, University of Silesia, Katowice, Poland
**3** SMCEBI, Chorzow, Poland

[*] nbailey@ruc.dk

## Abstract

We investigate the variation of the driving force for crystallization of a supercooled liquid along isomorphs, curves along which structure and dynamics are invariant. The variation is weak, and can be predicted accurately for the Lennard-Jones fluid using a recently developed formalism and data at a reference temperature. More general analysis allows interpretation of experimental data for molecular liquids such as dimethyl phthalate and indomethacin, and suggests that the isomorph scaling exponent $\gamma$ in these cases is an increasing function of density, although this cannot be seen in measurements of viscosity or relaxation time.



## 1   Introduction

Some years ago the glass group at Roskilde University identified a class of liquids which we have proposed should be considered "simple liquids" [1–8]. To make a distinction from conventional notions of liquid simplicity we refer to them as R (Roskilde)-simple liquids (or R-simple *systems*—the concept is not specific to the liquid phase). R-simplicity is characterized by how a typical configuration's potential energy changes under a uniform scaling of coordinates: if any two configurations with equal potential energy at one density still have equal energies when scaled to a new density, then the system is R-simple [9]. This definition highlights a scale invariance which is typically hidden [10]–that is, not obvious from the Hamiltonian. From it a large number of interesting consequences follow, the most important being the existence of *isomorphs*: curves in the phase diagram along which structure and dynamics are invariant when expressed in thermodynamically *reduced units*. Only systems with inverse power law potentials are perfectly R-simple, but to a very good approximation all van der Waals bonded systems and most metals [11] are R-simple, i.e. have good isomorphs. The theoretical development has mostly been validated by computer simulations, but there has also been experimental confirmation of specific predictions of isomorph theory [12, 13]. While most of the published work on isomorph theory is by members of our own group, broader interest has grown steadily: examples include high-order analysis of structure in simulated glass-forming liquids [14], simulations of bulk metallic glass [15], and a theoretical argument based on the infinite-dimensional limit [16]. The basic concepts have also earned a page in the most recent edition of Hansen and McDonalds's book on liquid theory [17]. The aim of this paper is to address the question: what are the consequences of isomorphs for crystallization kinetics? In particular, since along an isomorph liquid dynamics are invariant, we focus on the variation of thermodynamic driving force for crystallization. In the following subsections we give a fairly complete introduction to isomorph theory and discuss its consequences for simple theoretical models for crystal nucleation and and growth, before giving an outline of the remainder of the paper.

### 1.1   Features of R-simple systems

The most direct experimental consequence of isomorphs emerges naturally from the theory but was observed experimentally before its formulation: so-called *density scaling* of dynamics [18–25]. Having measured a liquid's relaxation time $\tau_\alpha$, for example using dielectric spectroscopy, over a range of temperatures and pressures, one plots the data as a function of $TV^\gamma$ where $V$ is the molar volume, or equivalently of $\rho^\gamma/T \equiv \Gamma$ where $\rho$ is the density; this collapses the data to a single curve in many cases. The density-scaling exponent $\gamma$ is an empirically determined material-specific parameter. Isomorph theory explains this collapse, while specifying the phenomenon more precisely: the collapse works better for certain liquids, those which are R-simple; the best collapse is obtained when plotting the relaxation time in *reduced units*, $\tilde{\tau}_\alpha \equiv \tau_\alpha \rho^{1/3} T^{1/2}$, the parameter $\gamma$ can depend on density, and it can be independently deter-

mined at a given state point from the liquid's dynamic response functions [12]. In general the scaling parameter is written $\Gamma = h(\rho)/T$ where $h(\rho)$ is the density scaling function, whose logarithmic derivative is $\gamma(\rho)$. If the latter is constant then we have the case of power-law density scaling $h(\rho) = \rho^\gamma$; theoretical models featuring strictly constant $\gamma$ are those with inverse power law (IPL) interactions. Non-constant $\gamma$ can be observed in simulations where large density changes can be considered, but is hard to observe experimentally [26–28]. Dynamical measurements and empirical density scaling allow isomorphs to be conveniently identified experimentally as *isochrones*, curves of constant (reduced) relaxation time.

A large part of the work on R-simple systems so far has focussed on identifying which quantities are expected to be invariant on isomorphs (when expressed in reduced units), and which are are not, and checking the invariances for different systems. The invariant quantities include measures of structure and dynamics, some thermodynamic properties and some transport coefficients [29]. Invariant thermodynamic properties include the configurational or excess entropy $S_{\mathrm{ex}} = S - S_{\mathrm{id}}$ (where $S$ is the total entropy and $S_{\mathrm{id}}$ the ideal gas entropy at the same density and temperature) and the isochoric specific heat $C_V$ [30]. The more recent formulation of the theory recognizes that $S_{\mathrm{ex}}$ is particularly fundamental, and that isomorphs should in fact be *defined* as curves of constant $S_{\mathrm{ex}}$ (configurational adiabats) [9]; moreover the small variation of $C_V$ on isomorphs can be accounted for within the theory. Notably absent from the list of invariant quantities are the potential energy, the Helmholtz free energy and its volume derivatives, pressure and isothermal bulk modulus. To understand their absence consider the following generic expression [31] for the potential energy of an R-simple system (not the most general possible [9], but sufficient for the present work):

$$U(\mathbf{R}) = h(\rho)\Phi(\tilde{\mathbf{R}}) + N g(\rho). \tag{1}$$

Here tilde denotes reduced coordinates, which for the positions are defined by $\tilde{\mathbf{R}} \equiv \rho^{1/3}\mathbf{R}$ (similarly any quantity with dimension length). The reduced coordinates are thus the coordinates in units of the mean interparticle spacing. The function $h(\rho)$ is the density scaling function mentioned above, related to $\gamma(\rho)$ through:

$$\gamma(\rho) \equiv \frac{d \ln h(\rho)}{d \ln \rho}. \tag{2}$$

The last term in Eq. (1) involves an arbitrary function of density. In this formulation the scaling properties of R-simple systems follow from the fact that scaling the coordinates of a configuration does not change the reduced coordinates, so the function $\Phi$ is unaffected. The energy simply gets scaled by $h(\rho)$, which can be compensated by an equal temperature scaling [6], and shifted by $N g(\rho)$. The latter does not affect structure and dynamics[1], but it does contribute to the potential energy, pressure and the other non-scaling quantities mentioned above, hence we refer to it as the non-scaling term. These contributions do not scale with density the same way that those from the first term do, hence the non-invariance of the above-listed quantities.

## 1.2 Crystallization kinetics along isochrones

The motivation for the present work comes from recent experiments by Adrjanowicz *et al.* [32–36] which have aimed to separately determine the kinetic and thermodynamic factors controlling crystallization of supercooled liquids by fixing the liquid's relaxation time $\tau_\alpha$. Crystallization under pressure is of interest both scientifically and in the context of industrial applications, including materials science, food and pharmaceuticals. Understanding the effect

---

[1]This is strictly only true for constant volume, but typical measures of dynamics are independent of ensemble, since local processes are not sensitive to boundary fluctuations.

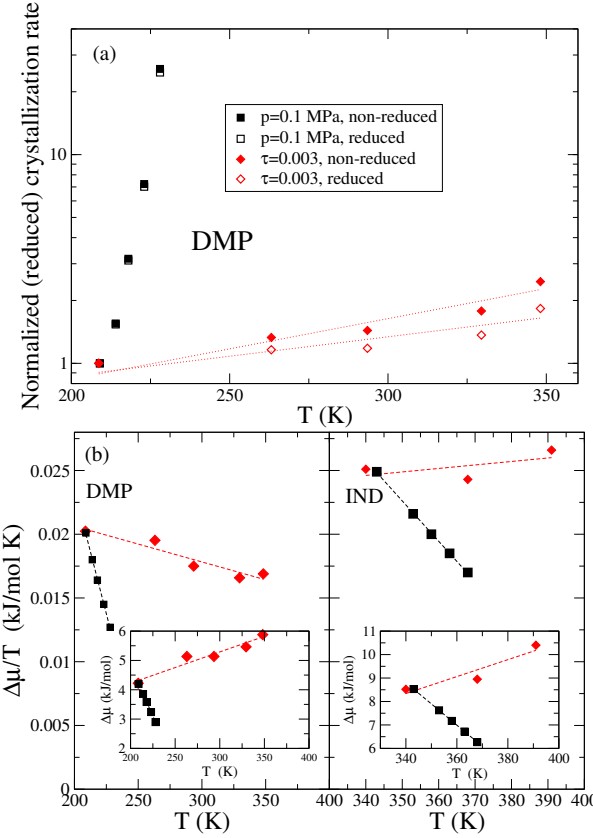

Figure 1: Revisiting isochronal crystallization data: (a) DMP crystallization rates [32], also in reduced units, on isochrone versus isobar. To compare trends all data, both non-reduced and reduced, have been normalized by the respective values for pressure 0.1 MPa, temperature 209 K. (b) Estimated driving force $\Delta\mu$ normalised by temperature (i.e. in reduced units) for (left) DMP, and (right) indomethacin [33]; in both cases the squares are data along the isobar $p = 0.1$MPa, while diamonds are data along the isochrone $\tau = 0.003$s. The inset shows the data without normalisation by temperature. Dashed lines are linear fits to show trends.

of pressure on crystallization theoretically is not straightforward, however, since the kinetic and thermodynamic factors respond oppositely: increased pressure, at constant temperature, generally slows a liquid's dynamics down, i.e. decreases the kinetic factor in the crystallization rate, while it increases the driving force (since high pressure favors the crystalline phase). The experiments of Adrjanowicz *et al.* sought to disentangle these effects by varying pressure, not at constant temperature, but while simultaneously varying temperature in order to keep $\tau_\alpha$ constant. In the language of the previous subsection, they measured along isochrones, albeit without taking reduced units into account. For the cases of indomethacin [33], a pharmaceutical, and dimethyl phthalate (DMP) [32], an increase in the crystallization rate $k$, was found, assumed due to an increase in the thermodynamic driving force $\Delta\mu$, since the kinetic factor was constant by construction. The increases in overall crystallization rates are significant, a factor of $\sim 4$ and $\sim 2$ for indomethacin and DMP, respectively. A separate analysis of thermodynamic data was used to estimate $\Delta\mu$ at the same state points on the isochrone. Consistent with the crystallization rate data, an increase with increasing density, was found both for indomethacin and DMP.

The observed crystallization kinetics quantified by the rate $k$ involve both nucleation and growth. In both processes, the driving force $\Delta\mu$ must play an essential role since neither can

proceed if it vanishes. The most common theoretical framework for describing nucleation rates is classical nucleation theory, according to which the rate involves a Boltzmann factor [37]

$$\exp\left(-\frac{16\pi\sigma^3}{3k_B T(\Delta\mu)^2}\right), \tag{3}$$

with $\sigma$ the interfacial free energy. While it is well known that CNT rarely gives accurate predictions for rates [38–40], and that it is not necessarily clear that the interfacial free energy is relevant or even well-defined when the critical nucleus is small, it is reasonable to assume that the nucleation rate depends on $\Delta\mu$ more or less as in Eq. (3). The first theoretical expression for the growth velocity, due to Wilson [41] and Frenkel [42], is given in [43] as

$$v = \frac{Da}{\Lambda^2} f \left(1 - \exp(-\Delta\mu/k_B T)\right), \tag{4}$$

where $a$ is the layer thickness, $D$ the self-diffusivity of the liquid, $\Lambda$ the mean free path and $f$ a factor accounting for the fraction of sites on the interface where growth actually occurs. This expression has the form of a kinetic factor multiplied by a thermodynamic one. Broughton *et al.* [43] found that in very simple systems, including the Lennard-Jones fluid, the diffusivity of the liquid plays no role, and the kinetic factor is simply the thermal velocity. They fitted Lennard-Jones data using the expression (modified slightly as in Ref. [44])

$$v = \frac{a}{\lambda}\sqrt{\frac{3k_B T}{m}} f \exp\left(-\frac{\Delta S}{k_B}\right)(1 - \exp(-\Delta\mu/k_B T)). \tag{5}$$

Here $\lambda$ is the distance an atom must move to join the crystal and $\Delta S$ is the entropy difference between liquid and crystal. In experiments and simulation both nucleation and growth can be more complex than implied by the classical models; for example the nucleated crystal may not be the stable phase, but rather an intermediate meta-stable crystal which is energetically more accessible from the melt. This is the Ostwald step rule [45], which has been observed in Lennard-Jones simulations [46].

A very natural question from the perspective of isomorph theory is whether crystallization rate is an isomorph invariant (in reduced units). If the rate is controlled by a kinetic factor, which is invariant, and the driving force, then the question becomes whether the driving force is invariant. There is experimental evidence that this is nearly true [35,36] but the question has not been addressed theoretically yet. The isomorph perspective immediately suggests alternative ways to formulate the question: first, the appropriate curve to consider is one of constant reduced relaxation time $\tilde{\tau}_\alpha = \tau_\alpha \rho^{1/3} T^{1/2}$; second, the measured quantities should also be expressed in reduced units. For example the reduced crystallization rate is $\tilde{k} = k\rho^{-1/3} T^{-1/2}$, while the reduced driving force is $\Delta\tilde{\mu} = \Delta\mu/T$. Third, the variation of these quantities along the isochrone does not necessarily tell the whole story. While factors of order 2-4 can seem substantial enough, order-of-magnitude changes appear when isotherms or isobars are considered. In that context it is potentially more interesting to investigate how close curves of constant driving force or curves of constant crystallization rate are to isochrones [35,36].

One can consider also the following observations: (1) free energies vary along isomorphs (also in reduced units), so one cannot immediately expect the reduced driving force to be invariant, although (2) *differences* in free energy could well be relatively invariant, if the non-scaling contribution is more or less the same for both phases and therefore cancels in the difference; (3) in particular, since the non-scaling term $g(\rho)$ depends only on density (at the level of approximation of Eq. (1)), the cancellation is likely to be more effective in systems where the density difference between liquid and crystal is relatively small. Without a detailed analysis, therefore, the best one can say *a priori* is that some variation of $\Delta\tilde{\mu}$ is to be expected, but it is likely to be small for ordinary liquids. A detailed analysis is the aim of this paper.

To round off this sub-section, in Fig. 1 (a) we present DMP data from Ref. [32] where the variation of crystallization rate along an isochrone (non-reduced) is compared to the variation along an isobar. Both $k$ and $\tilde{k}$ are plotted, and these are normalized by their values at the lowest temperature and pressure state point. While plotting $k$ in reduced units reduces the variation slightly, the main point is that either way the variation along the isochrone is small compared to that along the isobar. Note that the isochrone was determined using non-reduced units; there is no simple analysis that can account for that and demonstrate the variation along a reduced-time isochrone. In part (b) of the figure we show the estimated reduced driving force $\Delta\mu/T$ for both DMP and indomethacin, where the insets show the unreduced version of this quantity ($\Delta\mu$). The former is more relevant not just from an isomorph theory point of view but also in general theoretical expressions for the crystallization rate where a Boltzmann factor involving $\Delta\mu$ typically appears, see Eqs. (4) and (5). Indeed, along an isomorph, in reduced units, and assuming the parameters $a$ and $\Lambda$ scale as $\rho^{-1/3}$, Eq. (4) becomes

$$\tilde{v} = \tilde{M}\left(1 - \exp(-\Delta\tilde{\mu})\right), \tag{6}$$

where the dimensionless mobility $\tilde{M}$ is given by $\tilde{D}\tilde{a}f/\tilde{\Lambda}^2$ and is expected to be constant along an isomorph. Using Eq. (5) the dimensionless mobility becomes

$$\tilde{M} = \frac{\sqrt{3}af}{\lambda}\exp\left(-\frac{\Delta S}{k_B}\right); \tag{7}$$

unlike the Wilson-Frenkel expression, here the dimensionless mobility can vary slightly via $\Delta S$. Finally note that the Boltzmann factor in CNT in reduced units becomes

$$\exp\left(-16\pi\tilde{\sigma}^3/3(\Delta\tilde{\mu})^2\right); \tag{8}$$

the variation of $\Delta\mu$ will dominate as long as the reduced interfacial free energy can be assumed constant, but if both vary slightly then both are relevant.

Note that in Fig. 1 $\Delta\mu/T$ varies much less than $\Delta\mu$ by itself along isochrones. While for indomethacin it shows a weak increase, not much larger than the scatter in the data, for DMP it actually decreases slightly, which is not consistent with the increase in crystallization rate. The use of reduced units to define the isochrone would not change this apparent discrepancy: reduced isochrones have a higher slope in the $\rho, T$ plane than non-reduced ones (to compensate for the factor $\rho^{1/3}T^{1/2}$ in $\tilde{\tau}$, $\tau$ must decrease as $\rho$ and $T$ increase along the isochrone); for any given temperature this corresponds to moving closer to the freezing line, i.e. less supercooled, and therefore a reduction in $\Delta\tilde{\mu}$. Thus the use of reduced units to define the isochrone would decrease the slope of $\Delta\mu/T$ versus $T$. Since $\Delta\mu$ is not measured directly but estimated using extrapolated thermodynamic data, it is not surprising that errors in this procedure could convert a slight increase into a slight decrease. For the arguments presented in this work, we have chosen to trust the directly measured crystallization rate. Crystallization is a complex phenomenon, with the measured rate $k$ at a given state point involving both nucleation and growth processes, as well as the thermodynamic history of the sample. Nevertheless we assume for our analysis that variation of $k$ along isochrones can be ascribed to changes in crystal growth rate coming from changes in $\Delta\mu$. Note that the rapid increase of $k$ on the isobar is due to the kinetic factor whose variation far outweighs the decreasing $\Delta\tilde{\mu}$.

## 1.3 Recent developments of isomorph theory

Thinking about the driving force for crystallization has led to new insights into the thermodynamics of R-simple systems. These were recently applied to predicting the thermodynamics of melting and freezing [47], in particular for the Lennard-Jones model system, for which an analytical formalism based on the existence of isomorphs was developed and used to accurately

predict the shapes of the freezing and melting lines. Furthermore the variation of various thermodynamic and dynamic quantities along these lines was also predicted. The essential results from this work are (1) the new formalism which also allows basic thermodynamic quantities to be predicted along isomorphs, including the non-invariant ones about which little could be said before: pressure, free energy, bulk modulus; (2) the technique of using an isomorph as the zero-order basis of a perturbation-type calculation to make accurate calculations of thermodynamic properties. It should be noted that the new formalism is consistent with previous results: In particular previously derived expressions for the density dependence of $\gamma$ through the density scaling function $h(\rho)$ [30] and for the variation of the isochoric specific heat [9] can now be derived in a more unified way. Both the analytical formalism and the perturbation technique will be used in this work to analyze the variation of $\Delta\mu$ along liquid isomorphs.

### 1.4 Outline of the remainder of the paper

In this paper we consider a set of state points along a given liquid isomorph in the supercooled (or super-pressurized) regime. We are interested in the corresponding crystal phases at the same temperatures and pressures, and in particular the difference in chemical potential (Gibbs free energy per particle) between the liquid and the solid phase. With subscripts $l$ and $s$ indicating liquid and solid (crystal) phase respectively, we define the chemical potential difference as $\Delta\mu \equiv \mu_l - \mu_s$, which gives a positive number in the supercooled region of the phase diagram, corresponding to a positive driving force for crystallization. In the next section we derive a general expression for $\Delta\mu$ for an R-simple system. In section 3 we derive explicit expressions for the Lennard-Jones system and test them on simulation data. In section 4 we work with a more general, but less accurate formulation in order to analyze the experimental results. Finally this analysis is used to interpret the experimental data for DMP in Section 4.4 leading to the non-trivial result that its density scaling exponent $\gamma(\rho)$ must be an increasing function of density.

## 2 Thermodynamic driving force along a liquid isomorph: general theory

Recent theoretical developments [47] allow us to calculate pressures, free energies and more analytically along isomorphs for Lennard-Jones systems using simulation data at one point as input. One application is calculating the exact melting line using the isomorphs as the basis for a perturbation calculation [47]. The technique for calculating the crystallization driving force is almost identical. As with the melting line, the reference state point denotes a particular temperature and pressure. Because there are two densities it is convenient to consider temperature as the parameter along the isomorph(s), rather than densities. The treatment in this section is general, assuming only the existence of isomorphs described by $h(\rho)/T =$const., and not necessarily the validity of Eq. (1). We first summarize the procedure. We assume the same number of particles $N$ in either phase, thus we can compare the total Gibbs free energies; equivalently we could work with the Gibbs free energy per particle, i.e. the chemical potential $\mu$.

1. Choose a temperature $T$ and solve $h(\rho)/T = h(\rho_0)/T_0$ for $\rho$ to get the liquid and crystal densities on their respective isomorphs. Here $\rho_0, T_0$ refer to the reference state.

2. Calculate pressures and the (relative) Helmholtz free energies $F$ on both isomorphs.

3. Calculate the driving force as the liquid's Gibbs free energy $G$ on the isomorph minus the Gibbs free energy of the crystal at the same temperature and pressure. This requires a

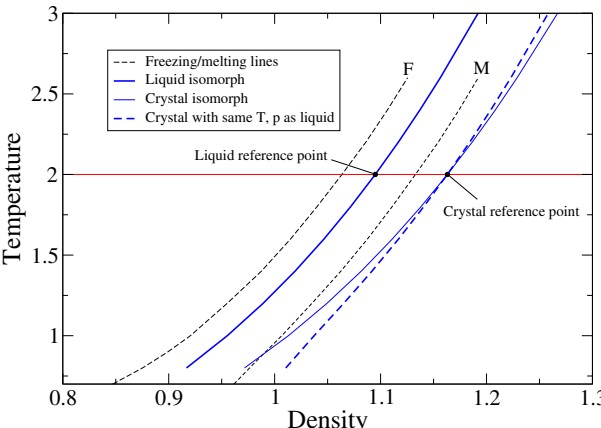

Figure 2: Illustration of how we calculate the driving force along a liquid isomorph (thick blue line) using data for the Lennard-Jones system. Notice that the liquid isomorph is to the right of the freezing line, that is, it is in the supercooled region. At a reference point on the liquid isomorph, specified by temperature $T = 2.0$, we know the crystal density which has the same pressure, and we know the driving force $\Delta\mu$. We construct a crystal isomorph through that point. At a different temperature isomorph theory tells us the free energies on both isomorphs, but for a given temperature the pressures are the two isomorphs are no longer equal. A Taylor expansion gives the correction to the crystal's free energy associated with fixing the pressure, and can also be used to find the corresponding crystal density (thick blue dashed line).

> Taylor expansion to correct for the difference in pressure between the liquid and crystal isomorphs.

The procedure is illustrated in Fig. 2. We now present the argument and procedure in more detail. Superscripts $I, l$ or $I, s$ indicates temperature-pressure points on an isomorph, the liquid or crystal one, respectively, while subscript $l$ or $s$ indicates the phase–because we are considering a non-equilibrium situation of a liquid from which a crystal is driven to nucleate and grow, we have two possible phases at a given thermodynamic state point. The notation allows us to label a quantity pertaining to the crystal whose temperature and pressure match those of a point on the liquid state isomorph. In terms of the Helmholtz free energy $F$, the $G$ of the liquid along the liquid isomorph is given by

$$G_l^{I,l} = F_l^{I,l}(T) + p^{I,l}(T)V_l^{I,l}(T). \tag{9}$$

We are interested in the crystal Gibbs free energy at the same temperature and pressure. Given that we know this at a reference temperature $T_0$, we can consider the crystal isomorph which includes the reference temperature and density. Along this isomorph the Gibbs free energy is

$$G_s^{I,s} = F_s^{I,s}(T) + p^{I,s}(T)V_s^{I,s}(T). \tag{10}$$

At a given temperature the pressure for the solid phase along the solid isomorph will not be equal to the pressure for the liquid phase along the liquid isomorph, even though they are equal at $T_0$.[2] On the other hand we expect the difference in pressure, $\Delta p \equiv p^{I,l}(T) - p^{I,s}(T)$, to be small in a system with good isomorphs. We need to move off the crystal isomorph at constant temperature to find the crystal state point which has the same pressure as the liquid— the crystal that eventually appears if the system is held at fixed temperature and pressure. We

---

[2]They are equal for systems with an inverse power law interaction, but not in general.

now make a Taylor expansion to first order in $\Delta p$, obtaining the desired crystal Gibbs free energy at a state point on the liquid isomorph as (for clarity we omit the $T$-dependence in the notation)

$$G_s^{I,l} = G_s^{I,s} + \left(\frac{\partial G_s}{\partial p}\right)_T^{I,s} \Delta p = G_s^{I,s} + V_s^{I,s}\Delta p, \tag{11}$$

where $V_s^{I,s}$ is the volume of the crystal phase on the crystal isomorph. Note again that when we refer to the crystal phase on the liquid isomorph we mean the crystal phase with the same temperature and pressure as the liquid phase on the latter's isomorph. In terms of the Helmholtz free energy we have

$$G_s^{I,l} = F_s^{I,s} + p^{I,s}V_s^{I,s} + V_s^{I,s}(p^{I,l} - p^{I,s}) \tag{12}$$

$$= F_s^{I,s} + V_s^{I,s}p^{I,l}. \tag{13}$$

Hence the desired free energy difference is given by

$$N\Delta\mu = G_l^{I,l} - G_s^{I,l} \tag{14}$$

$$= F_l^{I,l} - F_s^{I,s} + p^{I,l}V_l^{I,l} - V_s^{I,s}p^{I,l} \tag{15}$$

$$= F_l^{I,l} - F_s^{I,s} + p^{I,l}(V_l^{I,l} - V_s^{I,s}). \tag{16}$$

It is important to note that all quantities here are on isomorphs connected to the reference states. Going to second order in the Taylor expansion is also straightforward, see the end of Sec. 3 and Fig. 3 (b).

# 3 Driving force for Lennard-Jones systems

For the Lennard-Jones case, we can obtain an explicit expression for $\Delta\mu$ which can be evaluated using simulation data from a reference point. Eq. (1) is not used as more precise expressions are available for the Lennard-Jones case. The required input data at reference temperature $T_0$ consists of the pressure, density, potential energy, virial, and density scaling exponent $\gamma$ for both phases as well as $\Delta\mu(T_0)$. Starting from Eq. (16), we make further progress by separating the Helmholtz free energy into potential energy, configurational entropy and ideal gas contributions, the latter being [48]

$$F_{id} = Nk_BT\left(\ln\left(\rho\left(\frac{2\pi\hbar^2}{mk_BT}\right)^{3/2}\right) - 1\right). \tag{17}$$

This gives

$$N\Delta\mu = U_l^{I,l} - TS_{ex}^{I,l} + F_{id}(\rho_l^{I,l}, T) - U_s^{I,s} + TS_{ex}^{I,s} - F_{id}(\rho_s^{I,s}, T) + p^{I,l}(V_l^{I,l} - V_s^{I,s}) \tag{18}$$

$$= U_l^{I,l} - U_s^{I,s} - T\Delta S_{ex} + Nk_BT\ln(\rho_l^{I,l}/\rho_s^{I,s}) + p^{I,l}(V_l^{I,l} - V_s^{I,s}), \tag{19}$$

where $\Delta S_{ex}$ is the difference between liquid configurational entropy and crystal configurational entropy (on their respective isomorphs)–note that this is constant since isomorphs are by definition curves of constant $S_{ex}$; the logarithm comes from the difference in ideal gas contributions. All the quantities in this equation can be computed from their values at the reference temperature for Lennard-Jones systems, as explained below. The reference values are all straightforward to determine in simulations with one exception: the configurational entropy. Therefore we need also to know $\Delta\mu$ at $T_0$; then we can solve Eq. (19) for $\Delta S_{ex}$. In particular we have (with subscript 0 indicating the reference state point)

$$\Delta S_{ex} = \frac{1}{T_0}\left(-N\Delta\mu_0 + U_{l,0} - U_{s,0} + Nk_BT_0\ln(\rho_{l,0}/\rho_{s,0}) + p^{I,l}(V_l^{I,l}(T_0) - V_s^{I,s}(T_0))\right). \tag{20}$$

For Lennard-Jones potentials we use the expressions derived in Ref. [47], and reproduced in the appendix. The starting point is the key feature of isomorphs: that structure in reduced units is preserved along curves of constant excess entropy. For a term $(\sigma/r)^n$ in the Lennard-Jones potential (or its generalizations to arbitrary exponents), this means that its contribution to the potential energy varies like $\rho^{n/3}$ along an isomorph, with a coefficient which is a function of $S_{\text{ex}}$. For convenience we refer to reduced density $\tilde{\rho} \equiv \rho/\rho_{\text{ref}}$ where $\rho_{\text{ref}}$ is a reference density. Thus the potential energy for any state point can be expressed as

$$U = A_{12}(S_{\text{ex}})\tilde{\rho}^4 + A_6(S_{\text{ex}})\tilde{\rho}^2. \tag{21}$$

Using this expression as a starting point, by taking derivatives with respect to $\rho$ and $S_{\text{ex}}$ one can obtain expressions for all the basic thermodynamic quantities, involving in total six coefficients for each isomorph: $A_{12}$, $A_6$, $A'_{12}$, $A'_6$, $A''_{12}$ and $A''_6$ (for the first order approximation, Eq. (19), only the first four of these are necessary). The details are given in the appendix. In particular for the temperature we get (Eq. (34))

$$T = A'_{12}\tilde{\rho}^4 + A'_6\tilde{\rho}^2 = T_0\frac{h(\rho)}{h(\rho_{\text{ref}})}; \tag{22}$$

note that this is equivalent to the expression for $h(\rho)$ for Lennard-Jones systems derived in Ref. [30], with a slight difference in notation: their $h(\tilde{\rho})$ corresponds to our $h(\rho)/h(\rho_{\text{ref}})$. Inverting this equation allows us to solve for $\rho$ along each isomorph as a function of the main parameter $T$. The resulting densities can then be used in Eq. (21) and Eq. (33) to find the potential energy $U$ and virial $W$ on both isomorphs. Knowing $\rho$, $T$ and $W$ is enough to determine the pressure; thus all quantities in Eq. (19) are determined in terms of reference data.

Tests of Eq. (19) are shown in Fig. 3 (a) for several liquid isomorphs in the supercooled region of the Lennard-Jones phase diagram. The points at $T = 2$ are reference data; the predictions (black curves) must go through them by construction. The red squares represent independent determination of the driving force using the interface pinning method [49, 50], see appendix B for details. The driving force in plotted in reduced units, that is, normalized by the temperature, and shows a mild decrease with increasing temperature. The lowest curve was chosen to have $\Delta\mu = 0$ at the reference temperature, meaning that it intersects the freezing line there. The fact that $\Delta\mu$ deviates from zero away from the reference temperature is consistent with the freezing line not being an isomorph, although it is close to one [47,51]. Indeed the negative slope corresponds to the freezing line bending towards lower densities away from the reference isomorph, at temperatures lower than the reference temperature [49]. The mild decrease of $\Delta\tilde{\mu}$ is consistent with the reasoning in the introduction that for R-simple systems variation is expected to be small due to cancellation effects. The first order predictions, the black curves, are reasonably accurate for temperatures not too far from the reference, while small but significant deviations appear at low temperatures. These deviations could in principle stem either from a breakdown in the first order approximation we have made, or from a breakdown in the isomorph theory itself, meaning that we cannot trust the "zero-order" predictions, i.e. the expressions for the thermodynamic quantities along isomorphs. Indeed it is known that the isomorph predictions begin to fail at low temperatures in the vicinity of the triple point [47]. We can test for this by including second order corrections involving the square of the change in pressure from crystal to liquid isomorphs, i.e. adding a term $\frac{1}{2}(\partial^2 G_s/\partial p^2)_T(\Delta p)^2$ to Eq. (11). The second derivative of the Gibbs free energy with respect to pressure is essentially the isothermal bulk modulus, for which an expression involving the $A$ coefficients can easily be found (see appendix). The second-order predictions are plotted in Fig. 3 (b) and show a substantially improved match with the interface pinning data. This proves that it was not the isomorph theory that was at fault in Fig. 3 (a); we just needed to

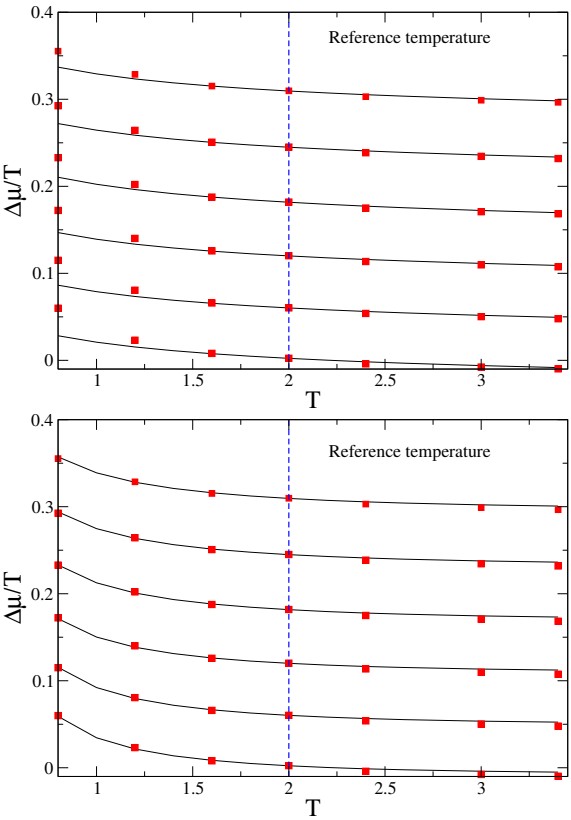

Figure 3: Reduced driving force along several liquid isomorphs, predictions based on data from the reference point $T_0 = 2.0$. Red squares were determined using thermodynamic integration and interface pinning [49, 50]. (a) first order approximation; (b) second-order terms added. The lowest curve corresponds to an isomorph which intersects the melting line at $T = 2.0$. The others are inside the supercooled region. The liquid densities at $T = 2.0$ are, from the bottom up, 1.065, 1.085, 1.105, 1.125, 1.145, 1.165.

go to higher order. Some very small systematic deviations seem to remain for $T > 2$ which we have not investigated further. The convergence to constant values at high temperatures is expected since in that limit we recover the limit of the $n = 12$ IPL coming from the repulsive term of the Lennard-Jones potential–all physical quantities are invariant in reduced units for IPL systems. It is not clear whether the IPL limit depends on exponent $n$; if not, then one would expect the value to be also valid for the hard-sphere case (choosing the packing fraction by equating excess entropies for example). Probably the values are relatively close, due to so-called *quasiuniversality* [31, 52]. We will next consider more general systems.

# 4  General R-simple systems

## 4.1  General isomorph equation of state

In this section we derive results for general R-simple systems with a view to being able to interpret experimental data for the crystallization driving force along isochrones. We start with the generic "perfect R-simple system" potential energy function given by Eq. (1). Note that this is not the most general R-simple potential energy function. In particular it assumes the scaling factor $h(\rho)$ depends only on density, whereas more generally it can depend also on

the potential energy itself (though not otherwise on microscopic coordinates) [9].

The form for the potential energy in Eq. (1) leads, via standard statistical mechanics (see Appendix C), to a Helmholtz free energy function

$$F(N, V, T) = N k_B T \phi(\Gamma) + N g(\rho) + F_{\text{id}}(N, V, T), \tag{23}$$

where $\Gamma \equiv h(\rho)/T$ is the scaling parameter which is constant on isomorphs and $F_{\text{id}}$ is the ideal gas contribution, Eq. (17). The factor $N$ (the number of particles) has been made explicit so that $\phi(\Gamma) \equiv -\ln(\int d^{3N}\tilde{R} \exp(-\Gamma \Phi(\tilde{\mathbf{R}}))/N)$ is an intensive quantity. There are three unknown functions in this expression: $h(\rho)$ (which also determines $\Gamma$), $\phi(\Gamma)$ and $g(\rho)$. We now consider more explicit forms for these, keeping them as simple as possible in order to make the reasoning clear.

For $h(\rho)$ the key question is whether it is the power law $\rho^\gamma$ (constant $\gamma$) or not (variable $\gamma$); an explicit expression for the latter case is not needed. For the invariant free energy $\phi(\Gamma)$, note that we are interested in small deviations from isomorphs. Thus changes of $\Gamma$ are expected to be small, so we can make a Taylor expansion about a reference value $\Gamma_{\text{ref}}$:

$$\phi(\Gamma) = \phi_0 + \phi_1(\Gamma - \Gamma_{\text{ref}}) + \frac{1}{2}\phi_2(\Gamma - \Gamma_{\text{ref}})^2. \tag{24}$$

For the non-scaling term, $g(\rho)$, we consider a simple power law form

$$g(\rho) = \lambda \rho^\alpha. \tag{25}$$

The exponent $\alpha$ must be different from $\gamma$, otherwise this term would be proportional to $h(\rho)$ and could be absorbed into the scaling term (essentially by adding the constant $\lambda$ to the function $\phi(\Gamma)$). The effect of this term, for example on the pressure variation along isomorphs, depends on whether $\alpha$ is greater than or less than $\gamma$, and on the sign of $\lambda$. One might expect physically a negative sign for $\lambda$, representing the attractive part of the potential. One must be careful when interpreting $g(\rho)$, however, because it is not uniquely defined—Eq. (23) could be rewritten with an arbitrary constant added to $\phi(\Gamma)$ and a corresponding multiple of $h(\rho)$ subtracted from $g(\rho)$, resulting in an equally valid decomposition of the free energy into scaling and non-scaling terms. A specific choice of functional form for $g(\rho)$ removes most of the ambiguity though.

For analyzing the consequences of the above forms for free energy for the driving force for crystallization, we make the additional assumption that the functions $h(\rho)$ and $g(\rho)$ are the same for both solid and liquid phases. We know from the Lennard-Jones system that there are in fact small differences, arising from the fact that isomorph theory is not an exact description, but these should not be too important; we are not attempting here to achieve the accuracy obtained in the Lennard-Jones case. The differences between liquid and crystal thus appear only in different values of the coefficients $\phi_0$, $\phi_1$ and $\phi_2$.

## 4.2 Pressure in general isomorph case

We start our investigation of the consequences of the decomposition (23) by considering the pressure, obtaining by differentiating the free energy with respect to volume:

$$p = \rho k_B T + k_B T \frac{d\phi}{d\Gamma} \rho \frac{d\Gamma}{d\ln\rho} + \rho^2 g'(\rho), \tag{26}$$

where the first term is the ideal gas contribution and $g'$ is the derivative of $g(\rho)$. This can be considered a reasonably generic equation of state for R-simple systems (not the most general since it does not allow for $C_V$ to vary along an isomorph [9]). Using the Taylor expansion (24)

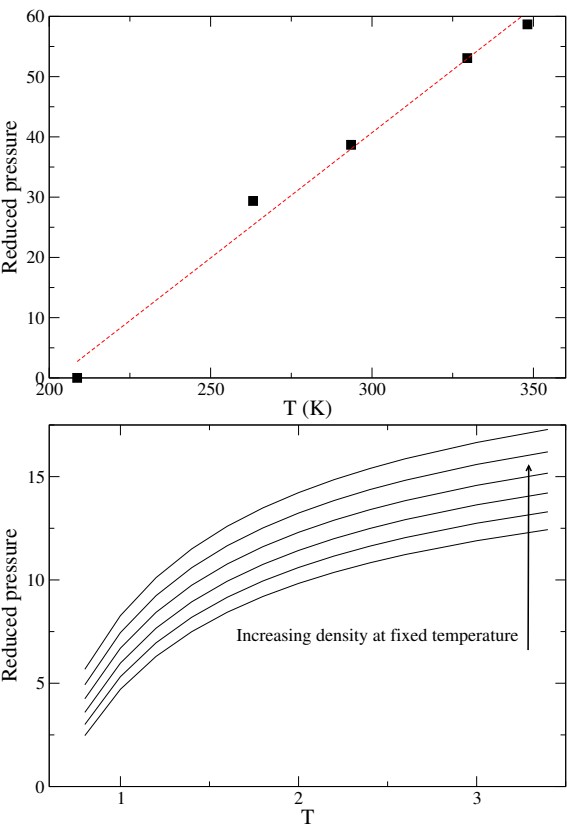

Figure 4: Left panel, reduced pressure along the $\tau = 0.003s$ isochrone for DMP. Right panel, reduced pressure along several isomorphs for the Lennard-Jones supercooled liquid. From bottom to top their densities at $T = 2.0$ are 1.065, 1.085, 1.105, 1.125, 1.145, 1.165. The variation of the reduced pressure along isomorphs gives information about the non-scaling part of the free energy. Note that the numerical values, although they are dimensionless, cannot be directly compared between systems since DMP is molecular, and the number density of molecules was used, while LJ is atomic. The fact that the experimental data show near-linear variation is likely due to the relatively small density range.

for $\phi(\Gamma)$, valid near a given reference isomorph, the identity $\frac{d\Gamma}{d\ln\rho} = \Gamma\gamma(\rho)$, and Eq. (25), we have

$$p = \rho k_B T + \rho k_B T \left(\phi_1 + \phi_2(\Gamma - \Gamma_{\mathrm{ref}})\right)\Gamma\gamma(\rho) + \lambda\alpha\rho^{\alpha+1}. \tag{27}$$

Taking the case of the pressure along the reference isomorph (i.e. $\Gamma = \Gamma_{\mathrm{ref}}$) we get, converting to reduced units,

$$\tilde{p} \equiv \frac{p}{\rho k_B T} = 1 + \phi_1 \Gamma_{\mathrm{ref}}\gamma(\rho) + \frac{\lambda\alpha\rho^{\alpha}}{T}. \tag{28}$$

From this we can see that variations in reduced pressure along an isomorph come from (1) variation in $\gamma$, which is expected to be small, certainly in experimental conditions, and (2) the non-scaling term. Assuming $\alpha < \gamma$ and $\lambda < 0$ the last term is negative and decreasing in magnitude, i.e. it causes the reduced pressure to increase as a function of density along an isomorph. This is consistent with what is observed in both Lennard-Jones systems and with the experimental data for DMP, see Fig. 4. Note though that a positive $\lambda$ together $\alpha > \gamma$ would also give such behavior. We are therefore interested in predictions which do not assume a specific sign of $\lambda$ or size of $\alpha$ but rather can be related to the behavior of the reduced pressure, which is directly measureable.

### 4.3 $\Delta\mu$ in general isomorph case

Eq. (16) provides a general expression for $\Delta\mu$ for a a system with isomorphs involving a first-order Taylor expansion of the Gibbs free energy. It requires that we know how the Helmholtz free energy varies along the liquid and crystal isomorphs. Using Eq. (23), (24) and (25) for the Helmholtz free energy in Eq. (16) we obtain the following explicit expression for $\Delta\mu$ along a liquid isomorph, given in reduced units as:

$$\Delta\tilde{\mu} \equiv \frac{\Delta\mu}{k_B T} = \ln(\rho_l/\rho_s) + \phi_0^{(l)} - \phi_0^{(c)} + \frac{\lambda}{k_B T}\left(\rho_l^\alpha - \rho_s^\alpha\right) + \left(1 + \phi_1^{(l)}\Gamma^{(l)}\gamma(\rho_l) + \frac{\lambda\alpha\rho_l^\alpha}{k_B T}\right)(1 - \frac{\rho_l}{\rho_s}) \tag{29}$$

In this expression the densities are those of the liquid and crystal on their respective isomorphs. Some rearrangement allows to see more clearly where possible variation arises:

$$\Delta\tilde{\mu} = \ln(\rho_l/\rho_s) + \Delta\phi_0 + T_3 + T_4, \tag{30}$$

where the third term

$$T_3 = \left(1 + \phi_1^{(l)}\Gamma^{(l)}\gamma(\rho_l)\right)\left(1 - \frac{\rho_l}{\rho_s}\right), \tag{31}$$

and the fourth term

$$T_4 = \frac{\lambda}{k_B T}\left((1 + \alpha(1 - \frac{\rho_l}{\rho_s}))\rho_l^\alpha - \rho_s^\alpha\right) \simeq \frac{\lambda}{k_B T}\left(\rho_l^\alpha - \rho_s^\alpha\right). \tag{32}$$

To make sense of these expressions it is useful to consider the case of a system of particles interacting through an inverse power law (IPL) interaction, for which isomorphs are exact, $\gamma$ is strictly constant and all physical properties are isomorph invariant [5]. The first term above, the logarithm coming from the ideal gas contribution, is exactly constant for the IPL case. Moreover it is unaffected by the presence of $g(\rho)$ since that does not affect the scaling properties of single-phase isomorphs and therefore the way density varies as a function of temperature along isomorphs. Deviations from IPL scaling, i.e. non-constant $\gamma(\rho)$, can cause variations in the ratio $\rho_l/\rho_s$, but changes in the logarithm are expected to be very small, and only potentially relevant near the freezing line where $\Delta\mu \simeq 0$.

The second term, $\Delta\phi_0 = \phi_0^{(l)} - \phi_0^{(c)}$ is exactly constant, coming as it does from the scaling contribution to the free energies. Significant variation potentially comes therefore from the terms $T_3$ and $T_4$. The first of these, $T_3$, is constant for constant $\gamma(\rho)$; it is hard to say in general what happens otherwise, since the factor $1 - \rho_l/\rho_s$ will have the opposite tendency to $\gamma$ itself[3]. But for the argument below it is just necessary to note that $T_3$ is constant if $\gamma$ is.

$T_4$ contains the variation due to the non-scaling term $g(\rho)$. Since $\rho_s > \rho_l$, and assuming as before $\lambda < 0$ and $\alpha < \gamma$, $T_4$ is expected to be positive and decreasing in magnitude, that is decreasing as a function of (liquid) density (the extra term multiplying $\rho_l^\alpha$, namely $\alpha(1 - \rho_l/\rho_s)$ is expected to be small enough that it does not change the overall behavior of $T_4$). Therefore, assuming we can neglect variation of $\gamma$, the only contribution to changes in $\Delta\tilde{\mu}$ is from $T_4$, and the sign of the variation is *opposite* to that of the corresponding contribution to the reduced pressure: If the reduced pressure increases as a function of density along an isomorph, then reduced driving force decreases as a function of density along an isomorph, and vice versa.

For non-constant $\gamma$ it is hard to say a priori which of $T_3$ and $T_4$ vary the most. For both of them the variation should be smaller than that of the corresponding terms in the reduced pressure (Eq. (26)): $T_4$ involves more or less the difference between the crystal and liquid densities raised to the power $\alpha$, and this difference varies less in absolute terms than either

---

[3]For increasing $\gamma(\rho)$ the crystal isomorph will be steeper than the liquid one at a given temperature therefore the ratio $\rho_s/\rho_l$ will decrease as one follows both isomorphs while increasing temperature.

density itself, while $T_3$ includes a factor $1 - \rho_l/\rho_s$, typically of order 0.05-0.1 (the same factor, multiplied by $\alpha$ appears in the third term, but added to unity, where it makes a smaller difference, assuming $\alpha$ is not too large).

### 4.4   Comparison to experimental data for DMP

The expression (30) for $\Delta\tilde{\mu}$ allows us to make a clear prediction in the case of a constant $\gamma$, connecting the variation in reduced pressure to that of the reduced driving force. But when we look at the experimental data for DMP—choosing to favor the crystallization rate data in Fig. 1 (a), over the estimated $\Delta\mu$ shown in Fig. 1 (b) and assuming that it reflects mainly growth and thereby variation in $\Delta\tilde{\mu}$—we find that both the reduced pressure and the reduced driving force increase as a function of density. The only way this can be rationalized within the isomorph formalism presented here is by allowing the scaling exponent to vary, which makes the $T_3$ term above vary: in particular $\gamma$ must increase as a function of density. In that case both the non-scaling term and the variation of $\gamma$ cause the reduced pressure to increase, while they contribute oppositely to the variation in the driving force, and the variation of $\gamma$ dominates. Thus the results of the analysis in this section can be summarized as follows:

1. For constant $\gamma$, and $g(\rho) = 0$, $\Delta\tilde{\mu}$ is invariant.

2. When $g(\rho) \neq 0$ but still with constant $\gamma$, then variation of the driving force in reduced units is inversely correlated with the variation of the pressure in reduced units. But this does not appear consistent with experimental data.

3. Variation of $\gamma$ also affects $\Delta\tilde{\mu}$. In LJ $\gamma$ decreases but there is some evidence that it tends to *increase* in real molecular liquids. Within the formalism described here, we must conclude that this must also be true for DMP.

An increasing $\gamma$ with density, while opposite to what is seen in Lennard-Jones-type potentials, can be obtained by simply moving the Lennard-Jones potential outward in $r$, so that the repulsive part diverges at a finite separation rather than at $r = 0$; this is physically reasonable for molecules constructed out of smaller units [53], although it remains to be investigated in detail for molecular models. Experimental systems with decreasing $\gamma(\rho)$ should include metals, but density scaling data for metallic systems does not exist to the best of our knowledge.

## 5   Conclusion

We have considered variation of $\Delta\mu$, more specifically its reduced-unit counterpart defined by $\Delta\tilde{\mu} \equiv \Delta\mu/T$, along liquid isomorphs in so-called R-simple systems. The motivation for this is the more challenging question of whether the complex process of crystallization, involving two different phases, could be invariant along an isomorph of the liquid phase. Since the crystallization rate $k$ involves in principle many factors, for the theoretical analysis we focus on $\Delta\mu$; for comparison with experimental data we have chosen to interpret variation in the measured rate $k$ along an isochrone as showing the variation in $\Delta\tilde{\mu}$.

Theoretically it is not obvious immediately whether in an R-simple system $\Delta\tilde{\mu}$ should be invariant along isomorphs, but a straightforward application of the approach developed in Ref. [47] gives a formalism for quantifying $\Delta\tilde{\mu}$. Moreover detailed predictions can be made for the Lennard-Jones model system which are in very good agreement with simulation data. In this case $\Delta\tilde{\mu}$ is a weakly decaying function of increasing temperature along a liquid isomorph. The experimental data for dimethyl phthalate is somewhat inconsistent: the reduced crystallization rate increases weakly, while the reduced driving force $\Delta\tilde{\mu}$ decreases weakly,

with increasing temperature along the isomorph. Part of the point is that the variation is weak, and that determining whether it is increasing or decreasing is challenging. We have chosen to trust the data for the crystallization rate $k$, assuming that its variation along an isochrone gives direct information about that of $\Delta\tilde{\mu}$. For indomethacin both kinds of data indicate an increasing $\Delta\tilde{\mu}$. Analysis of a more general equation of state for systems with perfect isomorphs allows conclusions to be drawn from the experimental data. Specifically the crystallization rate data, when combined with the variation of reduced pressure on the isomorph, are inconsistent with the assumption of a fixed $\gamma$; in fact one can conclude that in fact $\gamma$ must be an increasing function of density. The take-home message for experimentalists is therefore that variation of $\gamma$ can be relevant and can be inferred even if not directly detectable in for example density scaling analysis of dynamic data.

To make further progress in connecting the formalism here to experimental data requires more data, both thermodynamic and dynamic. In particular, more precise experimental determination of $\Delta\mu$ would be welcome, and both $k$ and $\Delta\mu$ should be measured along reduced-unit isochrones $\tilde{\tau}$ =const. This should resolve the apparent contradiction between the DMP data in parts (a) and (b) of Fig. 1. Analysis of the pressure on several isomorphs (experimentally, isochrones) should provide some constraints on the parameters $\lambda$ and $\alpha$ entering into $g(\rho)$. Accurate measurements of thermodynamic response functions such as the isochoric specific heat $C_V$, the isothermal bulk modulus $K_T$ and the pressure coefficient $\beta_V = (\partial p/\partial T)_V$ should in principle help to fit the other parameters (for example the coefficients $\phi_1$ and $\phi_2$ for both phases) but accounting properly for the kinetic terms is non-trivial in molecular systems. Nevertheless it would be fruitful to explore more fully (1) the formulation of equations of state which are designed to take isomorph invariance explicitly into account, and (2) methods of analyzing experimental data in order to constrain the parameters as much as possible.

Finally it can be discussed whether the crystallization rate $k$ determined via dielectric spectroscopy represents primarily growth or whether nucleation also plays a role. If nucleation plays a role then one must also consider the surface energy and its isomorph variation. This, and the variation along isomorphs of the nucleation rate, are planned to be addressed in future theoretical work.

## Acknowledgements

We thank Jeppe Dyre for his suggestion to go to second order in the analytic expressions for the Lennard-Jones systems. URP acknowledges support from the Villum Foundation through YIP grant no. VKR-023455. KN wishes to acknowledge The Danish Council for Independent Research for supporting this work. KA acknowledges funding from the Polish Ministry of Science and Higher Education within "Inventus Plus" project (0001/IP3/2016/74).

## A   Analytic expressions for thermodynamic quantities along iso­morphs

We reproduce here some of the derivations in Ref. [47] for the Lennard-Jones system. Starting with Eq. (21) we differentiate with respect to $\ln\rho$ at constant $S_{\text{ex}}$ to get the virial,

$$W = \left(\frac{\partial U}{\partial \ln\rho}\right)_{S_{\text{ex}}} = 4A_{12}\tilde{\rho}^4 + 2A_6\tilde{\rho}^2, \tag{33}$$

while differentiating Eq. (21) with respect to $S_{\text{ex}}$ gives the temperature

$$T = \left(\frac{\partial U}{\partial S_{\text{ex}}}\right)_\rho = A'_{12}\tilde{\rho}^4 + A'_6\tilde{\rho}^2, \tag{34}$$

where a prime indicates differentiation of $A_m$ with respect to $S_{\text{ex}}$. This expression is equivalent to $h(\rho)$ in the general formulation of isomorphs in the main text. Differentiating the last equation with respect to $\ln\rho$ gives the variation of temperature along an isomorph (i.e. at fixed $S_{\text{ex}}$):

$$\gamma T = \left(\frac{\partial T}{\partial \ln\rho}\right)_{S_{\text{ex}}} = 4A'_{12}\tilde{\rho}^4 + 2A'_6\tilde{\rho}^2. \tag{35}$$

For a given isomorph we can consider the $A$ coefficients and their first derivatives as numbers rather than functions. These parameters can be determined by considering a reference point on the isomorph with density $\rho_{\text{ref}}$ (so corresponding to $\tilde{\rho} = 1$). Denoting quantities measured in simulations at the reference state point with a subscript 0, we obtain two sets of two equations with two unknowns

$$U_0 = A_{12} + A_6 \tag{36}$$
$$W_0 = 4A_{12} + 2A_6 \tag{37}$$

and

$$T_0 = A'_{12} + A'_6 \tag{38}$$
$$\gamma_0 T_0 = 4A'_{12} + 2A'_6 \tag{39}$$

Solving these equations gives explicit expressions for the coefficients

$$A_{12} = (W_0 - 2U_0)/2 \tag{40}$$
$$A_6 = (4U_0 - W_0)/2 \tag{41}$$
$$A'_{12} = T_0(\gamma_0 - 2)/2 \tag{42}$$
$$A'_6 = T_0(4 - \gamma_0)/2. \tag{43}$$

Note that the expressions for $A'_{12}$ and $A'_6$ correspond to the previously published expression for $h(\rho)/h(\rho_{\text{ref}})$[4] for the Lennard-Jones potential [30]. These four coefficients are sufficient to both construct the isomorph ($T(\rho)$) and to calculate $U$, $W$ along it. From these the pressure is also readily obtained. For some quantities, such as the isochoric specific heat and the isothermal bulk modulus we need the second derivatives of the $A$ coefficients. We skip the intermediate steps here and quote the expressions (see Ref. [47]):

$$A''_{12} = (B_0 - 2T_0/C_{V,0}^{\text{ex}})/2 \tag{44}$$
$$A''_6 = (4T_0/C_{V,0}^{\text{ex}} - B_0)/2, \tag{45}$$

where we define the thermodynamic quantity $B$ as the derivative

$$B \equiv \left(\frac{\partial(T/C_V^{\text{ex}})}{\partial \ln\rho}\right)_{S_{\text{ex}}} = \left(\frac{\gamma T}{C_V^{\text{ex}}}\right)\left[1 + \left(\frac{\partial \ln\gamma}{\partial \ln T}\right)\right]. \tag{46}$$

The logarithmic derivative of $\gamma$ with respect to temperature appearing in the last expression can be calculated using either a finite difference with a nearby temperature, or a fluctuation expression [53]. These formulas refer to a given phase; determination of $\Delta\mu$ requires two

---

[4]Denoted $h(\tilde{\rho})$ in Ref. [30] due to their slightly different definition of $h(\rho)$ and use of unity as refence density.

Table 1: Coefficients in fifth degree polynomial representations of the computed chemical potential between solid and liquid along the isotherms $T = \{0.8, 1.2, 1.6, 2.0, 2.4, 3.0, 3.4\}$.

|  | $c_5$ | $c_4$ | $c_3$ | $c_2$ | $c_1$ | $c_0$ |
|---|---|---|---|---|---|---|
| 0.8 | $-1.1938\times10^{-5}$ | $2.7615\times10^{-4}$ | $-2.9270\times10^{-3}$ | $2.0803\times10^{-2}$ | $-0.15989$ | $0.18275$ |
| 1.2 | $-3.0894\times10^{-8}$ | $3.9765\times10^{-6}$ | $2.0183\times10^{-4}$ | $5.8332\times10^{-3}$ | $-0.13710$ | $0.72842$ |
| 1.6 | $1.1902\times10^{-8}$ | $-1.0382\times10^{-6}$ | $1.9095\times10^{-5}$ | $1.1924\times10^{-3}$ | $-9.9986\times10^{-2}$ | $1.1179$ |
| 2.0 | $6.6991\times10^{-9}$ | $-9.4839\times10^{-7}$ | $4.7095\times10^{-5}$ | $-4.5338\times10^{-4}$ | $-7.1807\times10^{-2}$ | $1.4206$ |
| 2.4 | $7.1340\times10^{-10}$ | $-1.4425\times10^{-7}$ | $8.9532\times10^{-6}$ | $1.7960\times10^{-4}$ | $-7.3224\times10^{-2}$ | $1.8371$ |
| 3.0 | $-1.0109\times10^{-9}$ | $2.7515\times10^{-7}$ | $-3.0572\times10^{-5}$ | $1.9434\times10^{-3}$ | $-0.11363$ | $2.8781$ |
| 3.4 | $-7.9384\times10^{-10}$ | $2.8524\times10^{-7}$ | $-4.114\times10^{-5}$ | $3.1242\times10^{-3}$ | $-0.1660$ | $4.1457$ |

isomorphs, one for liquid and one for the crystal. These are constructed separately starting from respective reference densities $\rho_{l,0}$ and $\rho_{s,0}$ at the same reference temperature $T_0$.

For the second order corrections to $\Delta\mu$ we take the next term in the Taylor expansion (11), namely

$$\frac{1}{2}\left(\frac{\partial^2 G_s}{\partial p^2}\right)_T^{I,s}(\Delta p)^2 = \frac{1}{2}\frac{V}{K_{T,s}^{I,s}}(p^{I,l} - p^{I,s})^2, \tag{47}$$

where $K_{T,s}$ is the isothermal bulk modulus of the crystal phase, here to be evaluated along the crystal isomorph. The resulting correction to $\Delta\mu$ is then

$$-\frac{1}{2}\frac{1}{\rho_s K_{T,s}^{I,s}}(p^{I,l} - p^{I,s})^2; \tag{48}$$

notice that here we also need the pressure of the crystal along its isomorph, which had otherwise cancelled out in the first-order expression. We take the following general expression for $K_T$

$$K_T = p + \rho\left(\frac{\partial W/N}{\partial \ln \rho}\right)_T, \tag{49}$$

and the expression given in Ref. [47] for the derivative of the virial with respect to $\ln\rho$,

$$\left(\frac{\partial W}{\partial \ln \rho}\right)_T = 4^2 A_{12}\tilde{\rho}^4 + 2^2 A_6\tilde{\rho}^2 - \frac{(4A'_{12}\tilde{\rho}^4 + 2A'_6\tilde{\rho}^2)^2}{A''_{12}\tilde{\rho}^4 + A''_6\tilde{\rho}^2}. \tag{50}$$

Eqs. (48), (49) and (50) together give the second-order correction to $\Delta\mu$.

## B  Calculating $\Delta\mu$ in simulations

Differences in chemical potential between solid and liquid $\Delta\mu$ of the LJ system (Fig. 3) are computed by thermodynamic integration in the super-cooled regime. For this computation we computed the volumes of the solid $v_s(p)$ and liquid $v_l(p)$ along isotherms computed in the constant $NVT$ ensemble (Fig. 5). Volumes along isotherms are fitted to fourth degree polynomials in the pressure (lines in Fig. 5). The difference in chemical potential at a given super-cooled state-point is computed by the polynomial integration

$$\Delta\mu(p,T) = \int_{p_m(T)}^{p} v_s(p') - v_l(p')dp', \tag{51}$$

where $p_m(T)$ is the coexistence pressure computed with the interface pinning method [49]. The coefficients of the resulting fifth degree polynomials

$$\Delta\mu(p,T) = \sum_{n=0}^{5} c_n(T)p^n \tag{52}$$

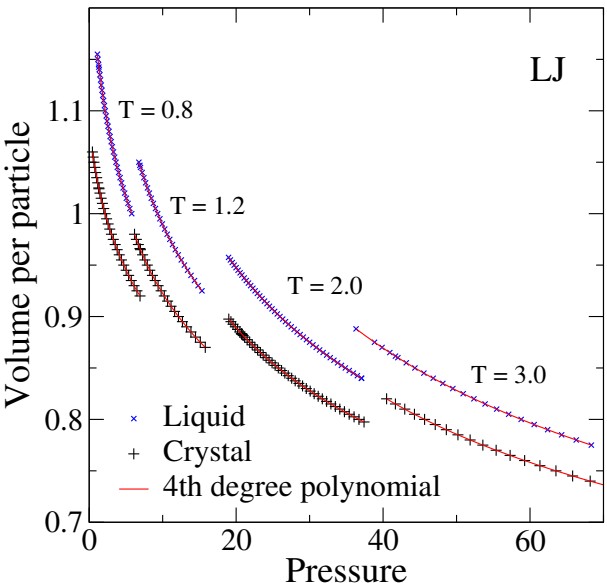

Figure 5: Volumes along the isotherms $T = \{0.8, 1.2, 2.0, 3.0\}$ of the LJ system in the solid and liquid state. 4th degree polynomial fits (lines) are subsequently used to compute the difference in chemical potential between solid and liquid $\Delta\mu$.

are given in Table 1.

## C  Scaling form of Helmholtz free energy

Here we supply some details for the decomposition of the Helmholtz free energy into scaling and non-scaling contributions used in subsection 4.1. In classical statistical mechanics we can write the total Helmholtz free energy $F$ as a sum of ideal gas and configurational terms $F = F_{\text{id}} + F_{\text{ex}}$, where [17]

$$F_{\text{ex}} \equiv -k_B T \ln \int \frac{d^{3N}R}{V^N} \exp\left(-U/k_B T\right) \tag{53}$$

$$= -k_B T \ln \int d^{3N}\tilde{R} \exp\left(-\Gamma\Phi(\tilde{\mathbf{R}}) - \frac{Ng(\rho)}{k_B T}\right). \tag{54}$$

Here the integral is over microstates. The non-scaling term does not depend on microstate so it comes out of the integral, giving

$$F_{\text{ex}} = -k_B T \ln \int d^{3N}\tilde{R} \exp\left(-\Gamma\Phi(\tilde{\mathbf{R}})\right) + Ng(\rho) \equiv Nk_B T\phi(\Gamma) + Ng(\rho). \tag{55}$$

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
