# Peer review of "Variation along liquid isomorphs of the driving force for crystallization"

_SciPost Physics, doi:SciPost Phys. 2, 022 (2017)_

## Round 1 · Referee Report · Anonymous (Referee 1) · 2017-2-22

Strengths

1- The article is written in a transparent, pedagogical way. 2- The article attempts to relate the perturbative approach of isomorphs to experimental measurements.

Weaknesses

1 - Some typos.

Report

In their work "Variation along liquid isomorphs of the driving force for crystallization" the authors illustrate a case study of application of the isomorphs theory to the calculation of the driving force to crystallisation for a simple liquid (Lennard-Jones fluid) and in the context of a selection of experimental results.

Proceeding from a previous work, [Pedersen, Ulf R., et al. "Thermodynamics of freezing and melting." Nature communications 7 (2016): 12386.] the authors discuss the details of a chemical potential calculation for a simple system, extend it to a class of isomorphic liquids and attempt a prediction for the experiments. From their discussion, it appears that the hypothesis of simple scaling does not apply to the considered experimental system, as the predictions are contradicted.

The article is well written and requires only minor modifications. There are, however, three aspects that could be explained by the authors more: 1- in Figure 3 (both left and right panel), the thermodynamic integration calculations appear to be overestimated for T>2 and underestimated for T<2 by the isomorphs theory. Can this be more explicitly rationalised? 2- related to the previous question, one could approximate the high temperature limit of Figure 3 by a suitably scaled system of hard spheres. How different would the estimate of the driving force be? Does the isomorph prediction converge to the hard sphere limit? 3-the discussion of the experimental data concludes, by exclusion, that $\gamma(\rho)$ should be a strong function of the density. What is the implication of this observation, in physical terms, on the nature of the interactions? Can the author propose a list of experimentally viable systems where they expect a milder dependence of $\gamma$ on the density, such that the predictions of the theory could be verified?

Requested changes

1-In the caption of Figure 2 a sentence is worded incorrectly: " but the for a given temperature...."
2-At page 10, the quantity "W" appears for the first time in the main text. Since it can be mistaken for other quantities, I suggest to modify the sentence as follows: "The resulting densities can be used in Eq.915) and Eq.(27) to find the $potential$ $energy$ U and the $virial$ W".
3-At page 14, the IPL (inverse power law) acronym appears without previous definition.

  • validity: high
  • significance: good
  • originality: high
  • clarity: high
  • formatting: excellent
  • grammar: excellent

Author:  Nicholas Bailey  on 2017-05-12  [id 133]

(in reply to Report 1 on 2017-02-22)

We thank the referee for careful reading of the manuscript and relevant feedback. We have made changes to the paper in response to both referees and will shortly resubmit. Here we detail the response to the points raised by the first referee.

$\bullet$ 1- in Figure 3 (both left and right panel), the thermodynamic integration calculations appear to be overestimated for $T>2$ and underestimated for $T<2$ by the isomorphs theory. Can this be more explicitly rationalized?

Reply: For the left panel the main deviations are due to the insufficient accuracy of the first order Taylor expansion. For the right panel the above statement is not true for all points, see the leftmost point on the topmost curve, for example. We have not investigated the remaining discrepancies, being satisfied at the agreement achieved by going to second order. It is true that there appears to be a systematic overestimation for $T>2$. Statistical uncertainty in the quantities determined by simulation at $T=2$ could possibly account for such a deviation in any one curve, but then we would expect the deviation to have the opposite sign in some of the curves.

Changes: We have added a comment in the discussion of Figure 3 mentioning the residual discrepancy.

$\bullet$ 2- related to the previous question, one could approximate the high temperature limit of Figure 3 by a suitably scaled system of hard spheres. How different would the estimate of the driving force be? Does the isomorph prediction converge to the hard sphere limit?

Reply: The referee raises a very interesting question. The values of $\Delta \mu$ in Figure 3 do converge to a fixed value for each isomorph, which is expected since we recover the $n=12$ IPL limit in the LJ system when increasing density and temperature along an isomorph (and all physical quantities are invariant in reduced units for IPL systems). As with IPL systems, the hard sphere phase diagram is one-dimensional, and one can in principle map each isomorph to a corresponding HS packing fraction. The most natural way from a fundamental point of view would be via the excess entropy, i.e. choose the packing fraction for which the excess entropy is the same as that along a given isomorph. An accurate expression for the value of $\Delta \mu$ as a function of the HS fluid packing fraction has been given by\cite{Ackerson/Schatzel:1995} (see also \cite{Auer/Frenkel:2001b}). Determining the excess entropy is computationally challenging, however, so we are not in a position to check this right now. We guess that for a given value of $S_{ex}$ the IPL values probably do not depend strongly on $n$, and since they must converge to the HS case as $n\rightarrow\infty$, presumably the limiting value on the LJ system is close to the HS value, but not necessarily equal.

Changes: Discussion of the above point has been added to the end of Section 3.

$\bullet$ 3-the discussion of the experimental data concludes, by exclusion, that $\gamma(\rho)$ should be a strong function of the density. What is the implication of this observation, in physical terms, on the nature of the interactions? Can the author propose a list of experimentally viable systems where they expect a milder dependence of γ on the density, such that the predictions of the theory could be verified?

Reply: We would not say that the conclusion is that $\gamma(\rho)$ is a "strong" function of density---the dependence is weak enough that it cannot be seen directly in dynamic data, for example. It is more that that looking at the variation of $\Delta \mu$ is a way to see that it varies, and that it is an increasing function of density. It is true that the variation has to be strong enough to outweigh the effect of $g(\rho)$ on $\Delta \mu$. Indeed, if $\gamma$ had been a decreasing function of density we wouldn't be able to infer it from our analysis, because then the effects of the background $g(\rho)$ term and the varying $\gamma$ would contribute the same way (corresponding to how the reduced pressure varies). Experimental systems where $\gamma(\rho)$ is expected to be decreasing include metals (metallic glass-forming liquids). Having a $\gamma$ that increases with density is consistent with interactions involving particles of finite volume, for example where a LJ-like potential diverges not at zero separation but at some finite separation (see \cite{Bailey/others:2013}) which is consistent with the interacting particles being molecules rather than atoms. There is a lot of work to be done on determining $\gamma$ and its density dependence for realistic molecular liquids.

Changes: Added to the end of Section 4: "An increasing $\gamma$ with density, while opposite to what is seen in Lennard-Jones-type potentials, can be obtained by simply moving the Lennard-Jones potential outward in $r$, so that the repulsive part diverges at a finite separation rather than at $r=0$; this is physically reasonable for molecules constructed out of smaller units \cite{Bailey/others:2013}, although it remains to be investigated in detail for molecular models. Experimental systems with decreasing $\gamma(\rho)$ should include metals, but density scaling data for metallic systems does not exist to the best of our knowledge."

The following specific textual issues were noted by the referee:

$\bullet$ 1-In the caption of Figure 2 a sentence is worded incorrectly: " but the for a given temperature...."

$\bullet$ 2-At page 10, the quantity "W" appears for the first time in the main text. Since it can be mistaken for other quantities, I suggest to modify the sentence as follows: "The resulting densities can be used in Eq.915) and Eq.(27) to find the potential
energy U and the virial W".

$\bullet$ 3-At page 14, the IPL (inverse power law) acronym appears without previous definition.

Reply/changes: We thank the referee again for careful reading of the manuscript. We have fixed the above issues.

---

## Round 1 · Referee Report · Anonymous (Referee 2) · 2017-3-10

Strengths

1) Novel approach to understand better how the crystallization dynamics depends on the state point.

Weaknesses

1) No discussion on the theoretical concepts used to describe crystallization 2) Basically no discussion on other numerical approaches to get the crystallization rate 3) No discussion of previous numerical results that predicated crystallization rate 4) The mechanism for crystallization is not really understood. In the present work the authors only calculate the difference in free energy. This is not enough to predict the crystallization rate 4) Introduction to isomorphs is way too long 5) Reference list is too strongly biased towards citations of the group of the authors. 6) The simplest system for crystallization is the hard sphere system. What can we learn from the present results for HS?

Report

Crystallization is not only an important process in many applications but
also a challenging scientific problem since it is not trivial at all to
come up with a reliable theoretical description for this phenomenon. The
most popular approach is probably the classical nucleation theory
(CNT), in which the crystallization rate is connected to the free
energy difference between the two phases and the diffusion constant of
the particles. Quite a few studies have been carried out to test this
idea (see, e.g. Auer and Frenkel 2001, Kawasaki and Tanaka 2010) but
none of these papers are cited. These studies have also shown that in
order to get the crystallization rate, it is not sufficient to know the
difference in the free energy between the two phases since the kinetics
plays also an important role. Furthermore these studies have also
shown that sometimes the liquid crystallizes by first forming (locally)
a phase that is different from the one of the target crystal, i.e. the free
energy difference between the liquid and the final crystal is not even
a good indicator for the driving force.

In the present manuscript the authors use the approach of the isomorphs
to calculate the free energy difference between the liquid and crystalline
phase. Unfortunately they do not explain at all why this is the relevant
quantity for the crystallization process since they do not even mention
CNT. They do discuss results of experiments in which the crystallization
dynamics has been studied as a function of temperature and pressure. But
at the end, see section 4.4, not much insight can be gained from the
present theoretical approach to rationalize the experimental findings.

More convincing are the direct comparison between the theoretical
results and computer simulations by means of which the authors that have
determined directly the free energy difference. The agreement between
theory and simulations is good once the second order correction terms
have been taken into account.

Further points:
-Is it really useful to give such a long introduction to the ideas of the
isomorphs?

-The authors cite 26 papers (out of 36 references) in which
they are coauthors or that are authored by members of their group. Doesn't
this indicate that either the citations are not balanced or that outside
this group nobody work on the approach with the isomorphs? There is,
e.g. the recent paper by Maimbour and Kurchan that have worked on this
subject as well. Also, Tolle (Rep. Prog. Phys. 2001) has studied the
dynamics of glass-formers as a function of pressure and temperature and
found that one can generate master curves.

p. 11, 3 lines after Eq. 17: What is the meaning of the sum over R?

In summary, it is to some extent interesting that the approach with the
isomorphs is able to give insight to the free energy difference between
the liquid and the crystalline phase. But as argued above, this does not
help much to advance our understanding of the crystallization dynamics.
Therefore I find it misleading that the authors use data from experiments
that probe the crystallization dynamics to motivate their calculations.
As it is the message of the paper is too confusing and hence I do not
recommend it to be accepted.

Requested changes

1) Make a much stronger point why this work is relevant for crystallization. If this is not possible refocus the message. 2) Add references 3) Take care of the items that are mentioned in the report

  • validity: ok
  • significance: low
  • originality: ok
  • clarity: ok
  • formatting: good
  • grammar: good

Author:  Nicholas Bailey  on 2017-05-12  [id 134]

(in reply to Report 2 on 2017-03-10)

We thank the referee for careful reading of the manuscript and relevant feedback. We have made changes to the paper in response to both referees and will shortly resubmit. Here we detail the response to the points raised by the second referee.

Overall the second referee's criticism appears to derive mainly from thinking that the paper aims to address the broad issue of understanding crystallization kinetics, because of the experiments which motivated our work. Therefore (s)he seems to expect more focus on the mechanistic details of crystallization. To clarify our message, we now mention already in the first paragraph that our focus is the following question: What are the consequences isomorphs for crystallization kinetics? Given that the kinetic factor is basically fixed along an isomorph/isochrone, the remaining dependence is expected to come from the driving force. The existence of isomorphs is analogous to a simplifying choice of coordinates in the phase diagram, such that the interesting physics is nearly independent of one coordinate. But of course we cannot claim to have understood anything about the dependence on the other coordinate (which would be any variable that uniquely labels isomorphs, such as the excess entropy), and this is where all complexities of crystallization actually come in. We address now the specific weaknesses listed by the second referee.

$\bullet$ 1) No discussion on the theoretical concepts used to describe crystallization

$\bullet$ 2) Basically no discussion on other numerical approaches to get the crystallization rate

$\bullet$ 3) No discussion of previous numerical results that predicated crystallization rate

Crystallization is not only an important process in many applications but
also a challenging scientific problem since it is not trivial at all to
come up with a reliable theoretical description for this phenomenon. The
most popular approach is probably the classical nucleation theory
(CNT), in which the crystallization rate is connected to the free
energy difference between the two phases and the diffusion constant of
the particles. Quite a few studies have been carried out to test this
idea (see, e.g. Auer and Frenkel 2001, Kawasaki and Tanaka 2010) but
none of these papers are cited. These studies have also shown that in
order to get the crystallization rate, it is not sufficient to know the
difference in the free energy between the two phases since the kinetics
plays also an important role. Furthermore these studies have also
shown that sometimes the liquid crystallizes by first forming (locally)
a phase that is different from the one of the target crystal, i.e. the free
energy difference between the liquid and the final crystal is not even
a good indicator for the driving force.

Reply: We agree with the reviewer that understanding crystallization kinetics is very challenging and that our work does not address all its complexities. We emphasize that our intention in this work is more restricted than the reviewer seems to think. Rather than trying to formulate a complete description of the mechanism, we focus on the driving force as one ingredient in crystallization kinetics, both in nucleation and growth. Other aspects will be considered in future work. The point of the experiments measuring crystallization rates on isochrones was to eliminate the kinetic factor, so that the variation of experimentally measured rate in this case can be associated with variation of the driving force alone. The promise of isomorph theory is to reduce the complexity of the (thermodynamic) parameter space by effectively reducing the dimensionality of the phase diagram by one. So for a pure system, in the relevant region the phase diagram is effectively one-dimensional. This is not exactly true however, since not everything is invariant along isomorphs, and this paper focuses on such residual variation. In this restricted context, the free energy difference is expected to play the largest role. Ongoing and future work will address the role of for example surface tension for nucleation, and the entropy difference between phases, in growth (both quantities are expected to vary only slightly along isomorphs). Regarding cases where a metastable crystal phase nucleates first (Ostwald step rule), isomorph theory can say that if this occurs in a system with good isomorphs, then its occurrence is almost equally likely along an isomorph, something that would be interesting to check.

Changes: We have added a new paragraph to Section 1.2 discussing the complexity of crystallization kinetics, including the basic expressions from CNT and the Wilson-Frenkel model for growth rate. For CNT we argue that even though it is not useful for accurate prediction of crystallization rates, it gives an idea of what quantities might be relevant. Since our paper is not an attempt to determine the crystallization rate as such, but rather to address its variation along an isomorph, we have not added a discussion of previous numerical methods and results but have cited several relevant works including those mentioned by the referee, while mentioning the limitations of classical models.

$\bullet$ 4) (a) The mechanism for crystallization is not really understood. In the present work the authors only calculate the difference in free energy. This is not enough to predict the crystallization rate

In the present manuscript the authors use the approach of the isomorphs
to calculate the free energy difference between the liquid and crystalline
phase. Unfortunately they do not explain at all why this is the relevant
quantity for the crystallization process since they do not even mention
CNT. They do discuss results of experiments in which the crystallization
dynamics has been studied as a function of temperature and pressure. But
at the end, see section 4.4, not much insight can be gained from the
present theoretical approach to rationalize the experimental findings.

Reply: We have not been clear enough about the motivation for studying the free energy difference. The point is to emphasize the simplicity brought by the existence of isomorphs: By measuring along an isomorph (or isochrone), which is by keeping the kinetic factor constant, only thermodynamic effects are relevant. Along an isomorph $\Delta \mu$ should be indeed be the dominant factor controlling crystallization kinetics, or at least growth kinetics. This follows explicitly from the theoretical expressions for growth rate both for diffusion-limited growth and diffusion-less growth exhibited by e.g. the Lennard-Jones system. For the nucleation rate, in the framework of CNT, the interface free energy also plays a role, if it varies along isomorphs (this is currently under investigation).

Changes: Related to the new paragraph on crystallization we show that along an isomorph the theoretical expressions become dependent primarily on $\Delta \mu$, under reasonable assumptions about the isomorph invariance of remaining factors.

$\bullet$ 4) (b) Introduction to isomorphs is way too long

-Is it really useful to give such a long introduction to the ideas of the
isomorphs?

Reply: We respectfully disagree with the referee that the introduction is too long. Since (1) isomorphs are a relatively new theoretical concept and (2) we wish to make the paper accessible to broad audience, including experimentalists, we feel a comprehensive introduction to isomorphs is warranted. The actual length is less than two pages (single column format) which does not seem excessive.

$\bullet$ 5) Reference list is too strongly biased towards citations of the group of the authors.

-The authors cite 26 papers (out of 36 references) in which
they are coauthors or that are authored by members of their group. Doesn't
this indicate that either the citations are not balanced or that outside
this group nobody work on the approach with the isomorphs? There is,
e.g. the recent paper by Maimbour and Kurchan that have worked on this
subject as well. Also, Tolle (Rep. Prog. Phys. 2001) has studied the
dynamics of glass-formers as a function of pressure and temperature and
found that one can generate master curves.

Reply: Since the concept of isomorphs has been mostly developed by our group, it is natural that a large fraction of the related cited work is from our group or involves collaborations with our group. Interest outside our group is steadily growing however. We have now listed some examples of this in the paper (including the Maimbourg and Kurchan work mentioned by the referee). The theory has also been mentioned in the latest edition of the "bible" of liquid theory, Theory of Simple Liquids by Hansen and McDonald. The referee mentions pioneering work by Tölle on what has become known as density scaling. We listed several of the early experimental works on density scaling but had omitted Tölle's. This has now been rectified. The ratio mentioned by the referee is now 26 out of 47 references.

Changes: We added the following towards the end of the first paragraph in the introduction:

"While most of the published work on isomorph theory is by members of our own group, broader interest has grown steadily; examples include high-order analysis of structure in simulated glass-forming liquids\cite{Malins/Eggers/Royall:2013}, simulations of bulk metallic glass\cite{Hu/others:2016}, and a theoretical argument based on the infinite-dimensional limit\cite{Maimbourg/Kurchan:2016}. The basic concepts have also earned a page in the most recent edition of Hansen and McDonalds's book on liquid theory\cite{Hansen/McDonald:2013}." We have added (Tölle et al., 1998) and (Tölle, 2001) to the list of references in second paragraph of introduction.

$\bullet$ 6) The simplest system for crystallization is the hard sphere system. What
can we learn from the present results for HS?

Reply: We refer to our reply to the first referee who also asked about the comparison to the hard-sphere system.

$\bullet$ p. 11, 3 lines after Eq. 17: What is the meaning of the sum over R?

Reply/changes: We noticed ourselves after submission that this was unclear. For one thing the expression was missing a logarithm (from converting the partition function to the free energy). This has been clarified in an additional appendix.

---

## Round 2 · Referee Report · Anonymous · 2017-5-25

Strengths

1 - Well written and presented
2 - The article deepens our knowledge of the isomorphs
3 - The authors pertinently responded to the referees' requests and improved their work further

Weaknesses

1 - The article is of moderate impact.

Report

The authors have implemented most of the relevant changes requested by the referees and also pertinently replied to the questions raised.

In particular, it appears to me that the 2017-3-10 report missed the general scope of the article which is not to to provide a complete novel theory of crystallisation that includes its thermodynamic and kinetic aspects. More modestly, the article provides an insight on the thermodynamic driving forces for crystallisation from the point of view of the theory of isomorphs.

The authors replied rather clearly to the reports and improved the form of the presentation so that their aims are now clearer.

I would recommend publication at this stage.

Requested changes

None.

---

## Round 2 · Author Response

We submit a revised version of our manuscript to SciPost Physics. We have made changes in several places in the paper in response to points raised by the reviewers. These consist mostly of additional text. The overall structure and the figures are unchanged. A few more changes are detailed below.

---

## Round 2 · List of Changes

1. The first subsection in section 1 starts now after the first paragraph, which now functions as a more overall introduction.
2. The term ``Roskilde (simple) liquid/system'' has been changeded to ``R-simple liquid/system'' to be consistent with current practice within the group.
3. A reference was added in the Introduction just before equation (1).
4. In appendix B we removed an unnecessary subscript on $\Delta \mu$.
5. A new appendix C has been added to clarify the derivation of Eq (22) (this was also requested by the second reviewer).
6. Minor re-writing of individual phrases and sentences.

---

## Editorial Decision

published